# A Study on Pharmacokinetics of Acetylsalicylic Acid Mini-Tablets in Healthy Adult Males—Comparison with the Powder Formulation

**DOI:** 10.3390/pharmaceutics15082079

**Published:** 2023-08-03

**Authors:** Noriko Hida, Taigi Yamazaki, Yoshiaki Fujita, Hidehiro Noda, Takehiko Sambe, Kakei Ryu, Takuya Mizukami, Sachiko Takenoshita, Naoki Uchida, Akihiro Nakamura, Tsutomu Harada

**Affiliations:** 1Division of Clinical Research and Development, Department of Clinical Pharmacy, School of Pharmacy, Showa University, Tokyo 142-8555, Japan; t.yamazaki@cmed.showa-u.ac.jp; 2Clinical Research Institute for Clinical Pharmacology and Therapeutics, Showa University, Tokyo 157-8577, Japan; t-sambe@med.showa-u.ac.jp (T.S.); ryu-k@med.showa-u.ac.jp (K.R.); mizukamit@med.showa-u.ac.jp (T.M.); s3take@med.showa-u.ac.jp (S.T.); nuchida@med.showa-u.ac.jp (N.U.); 3Division of Pharmaceutics, Department of Pharmacology, Toxicology and Therapeutics, School of Pharmacy, Showa University, Tokyo 142-8555, Japan; yoshiaki@pharm.showa-u.ac.jp (Y.F.); hironak@pharm.showa-u.ac.jp (A.N.); tharada@pharm.showa-u.ac.jp (T.H.); 4Pharmacy, Showa University Fujigaoka Hospital, Yokohama 227-8501, Japan; hidehiro@cmed.showa-u.ac.jp; 5Division of Clinical Pharmacology, Department of Pharmacology, School of Medicine, Showa University, Tokyo 142-8555, Japan

**Keywords:** acetylsalicylic acid, mini-tablet, powder, bioequivalence study, personalized medicine, hospital formulation, Kawasaki disease

## Abstract

Children with Kawasaki disease are prescribed acetylsalicylic acid powder as an antipyretic analgesic and antiplatelet agent; however, some of it remains in the mouth, leading to a bitter or sour taste. To address this issue, an in-hospital mini-tablet formulation of acetylsalicylic acid was developed. In order to use the mini-tablets safely and effectively, dissolution tests alone are not sufficient. Therefore, an open-label crossover study on six healthy participants was conducted to evaluate comparative pharmacokinetic parameters. The pharmacokinetic parameters of salicylic acid were C_max_: 4.80 ± 0.79 mg/L (powder; P), 5.03 ± 0.97 mg/L (mini-tablet; MT), AUC_0–12_: 18.0 ± 3.03 mg-h/L (P), 18.9 ± 4.59 mg-h/L (MT), those of acetylsalicylic acid C_max_: 0.50 ± 0.20 mg/L (P), 0.41 ± 0.24 mg/L (MT), AUC_0–12_: 0.71 ± 0.27 mg-h/L (P), 0.61 ± 0.36 mg-h/L (MT), with no significant differences between the mini-tablet and powder formulations. Although pharmacokinetic results obtained from adults cannot be directly applied to children, the results of this study are important for predicting pharmacokinetics. Furthermore, a formulation that can improve medication adherence in children who have difficulty taking acetylsalicylic acid powder, thus contributing to pediatric drug therapy.

## 1. Introduction

The mainstay of pediatric care is drug therapy. The World Health Organization (WHO) states that it is preferable to prescribe solid oral preparations for children [1]; however, for infants, syrups or fine granules suspended in water are often preferred to tablets or other solid preparations for oral drug administration. In Japan, 87% of all prescription drugs sold are oral drugs, in the following order: dry syrups, fine granules, tablets, and syrups [2]. The dosages and administration methods of dry syrups and fine granules are often different from those of tablets and syrups because the dosages and administration methods of these oral drugs can vary. Syrups and powder formulations are convenient for dosage adjustment and are easy to administer. However, taste, roughness, large volume, and odor often lead to refusal and difficulty in administration to children. Drug compliance is significantly reduced when a drug has an unpleasant taste or odor or is difficult to take [3].

The European Pediatric Translational Research Infrastructure assessment of children’s dosage form preferences in European countries found that the preferences of respondents (*n* = 1172) for oral dosage forms differ mainly by age, health status, and experience, with liquid formulations (35%) being the most preferred, followed by tablets (19%) and capsules (14%). They also found that granules were less preferred by adolescents (52.8%) [4]. Since Sarah et al. reported that mini-tablets are well-tolerated in children aged 2–6 years in 2009 [5], mini-tablets have garnered attention as a new pediatric oral solid dosage form. Oral liquid formulations have disadvantages, such as low stability, excipients that are potentially toxic to children, and poor transportability [6]. Multiple preparation steps and dose calculations are another risk factor for medication errors in children, and, subsequently, dosage and frequency of administration can lead to non-compliance [7]. In Japan, powder and fine granules are popular pediatric dosage forms [2]; however, their taste and roughness present difficulties [7]. Mini-tablets can circumvent problems associated with syrups, powders, and fine granules, making them easy to develop globally; however, studies on the dosing and acceptability of mini-tablets in pediatric patients have only been conducted using a placebo [8,9].

In clinical practice, children are sensitive to smell and taste and are often reluctant to take their medications. To improve adherence, pharmacists and other healthcare professionals collaborate to provide medication guidance [10]. To address the problem of children with Kawasaki disease having difficulty taking acetylsalicylic acid, mini-tablets with a diameter of 3 mm were developed from an acetylsalicylic acid powder approved by the Pharmaceutical Affairs Law. The acetylsalicylic acid mini-tablets used in this study were selected and adjusted by the Pharmacy Department of Showa University Hospital following the Guidelines of the Japan Hospital Pharmaceutical Association [11]. To administer mini-tablets containing medicinal ingredients to children in daily clinical practice, it is necessary to ensure quality and evaluate, in advance, the bioequivalence of the product with existing formulations [12]. Therefore, the purpose of this study was to investigate the pharmacokinetic parameters of acetylsalicylic acid powder and mini-tablets in six healthy adults. The results showed that the acetylsalicylic acid mini-tables had bioavailability comparable to that of acetylsalicylic acid powder. Although the results of pharmacokinetic parameters in adults are not directly applicable to children, they are important for predicting pharmacokinetics and drug efficacy. The outcome of studies conducted to assess the administration of the drug to children in real-world clinical practice is anticipated.

## 2. Materials and Methods

### 2.1. Dissolution Test

#### 2.1.1. Test Method

A dissolution study was conducted on acetylsalicylic acid mini-tablets and powder following the dissolution properties of acetylsalicylic acid in each article of the Japanese Pharmacopoeia.

Dissolution studies were conducted on acetylsalicylic acid mini-tablets and powder following the “Guidelines for Bioequivalence Studies of Generic Drugs” (24 November 2006, Pharmaceutical Affairs Bureau, Japan, No. 1124004).

Test formulation: Acetylsalicylic acid mini-tablets.The paddle method (50 rpm; JP 1st fluid, pH 1.2, 900 mL; 37 ± 0.5 °C) was employed according to the JP guidance using the dissolution tester NTR-6400A (Toyama Sangyo Co., Osaka, Japan). Aliquots were sampled at 5, 15, 30, 60, and 90 min using the autosampler SAS-6000 (Toyama Sangyo Co., Osaka, Japan). The dissolution profile similarity was determined following the bioequivalence guidelines for generic drugs based on the following criteria when the standard drug product did not dissolve an average of ≥85% of the standard drug product within 30 min: when the average dissolution rate of the standard formulation is ≥85% at the specified test time, the average dissolution rate of the test formulation is within ±15% of the average dissolution rate of the standard formulation at two appropriate time points around 40% and 85%, or the f2 function value is ≥42.

#### 2.1.2. Preparation and Storage of Acetylsalicylic Acid Powder

The following procedure was used to prepare and store the test drug (powder formulation):

Acetylsalicylic acid powder (100 mg) was weighed on an electronic balance, divided into portions using a desktop packaging machine, and then stored in a zippered aluminum bag with a desiccant in an investigational drug control cabinet with constant temperature and humidity.

#### 2.1.3. Preparation and Storage of Acetylsalicylic Acid Mini-Tablets

The following procedure was used to prepare mini-tablets:Acetylsalicylic acid powder (50 g) was measured and placed in a plastic bag #1.Crystalline cellulose (10 g), corn starch (10 g), and mannitol (27 g) were introduced to a separate plastic bag #2, and put into plastic bag #1 containing acetylsalicylic acid.The mouth of the plastic bag #1 was closed after allowing the entry of sufficient air. The bag was rotated 200 times manually to thoroughly mix the powders.Magnesium stearate (3 g) was added to plastic bag #1 containing the powder mixed with acetylsalicylic acid, crystal cellulose, and mannitol.The mouth of the plastic bag was closed after allowing the entry of sufficient air. The bag was rotated 200 times by hand to thoroughly mix the powders.The mixed powder was visually inspected for foreign substances; if any foreign substance was identified, they were removed, or a fresh batch was initiated.

A punch and die with a diameter of 3 mm were set in a single-shot tableting machine (N-30E, Okada Seiko Co., Ltd., Tokyo, Japan). The mixed ingredients were introduced continuously, and the tablets were compacted to a weight of 20 mg. Each tablet contains 10 mg of acetylsalicylic acid.

The tablets were placed on a sieve, vibrated to shake off the adhering powder, and temporarily stored with a desiccant in an aluminum bag with a zip-lock cover lid.Ten mini-tablets were counted and divided into pile packers. The tablets were stored in an aluminum bag with a zipper, together with a desiccant, in an investigational drug control room, with constant temperature and humidity.

### 2.2. Subjects

This clinical study was conducted at the Showa University Clinical Research Institute for Clinical Pharmacology and Therapeutics and was approved by the Clinical Research Review Committee of Showa University Educational Corporation (jRCTs031200140). The study included six participants who received explanations, consented, and confirmed their eligibility. The study was conducted from January to February 2021.

### 2.3. Eligibility

#### 2.3.1. Inclusion Criteria

Age: Participants must be ≥20 ≤45 years old when providing consent.Gender: Male.The ability to provide consent, comply with the study rules, undergo a preliminary examination as specified in the research protocol, and report any subjective symptoms.Deemed eligible for participation by the principal investigator or sub-investigator based on the preliminary examination outlined in the research protocol.

#### 2.3.2. Exclusion Criteria

People with a medical history that may affect the evaluation and safety of the study, including drug abuse/dependence, alcohol abuse/dependence, or heart, liver, kidney, lung, eye, and blood diseases.People with a history or current presence of peptic ulcer.People with a bleeding tendency.People with aspirin-induced asthma.People taking medications (including dietary supplements) that may affect the evaluation and safety of the study.People with a history of drug allergies.People with excessive alcohol intake (inability to maintain abstinence from alcohol intake during the study period).People who have participated in another clinical research study within the past three months.People deemed ineligible by the principal investigator or sub-investigator based on the preliminary examination specified in the research protocol.

### 2.4. Study Design

This study was conducted to investigate the dosing of the two drugs in two phases among six study participants whose blood concentrations were measured over time. Figure 1 shows the drug administration schedule, with the standard formulation (acetylsalicylic acid powder) administered to the study participants in the first period and the test formulation (acetylsalicylic acid mini-tablets) administered in the second period, both under open-label conditions. A washout period of 6 d was used for drug withdrawal between the two periods. Blood drug concentrations were measured according to the study schedule and are presented in Table 1.

This study was conducted following the Declaration of Helsinki, the Ethical Guidelines for Medical Research Involving Human Subjects, and the Clinical Research Act.

### 2.5. Study Formulations

#### 2.5.1. Drugs Administered

Acetylsalicylic acid powder (manufactured and distributed by Yoshida Pharmaceutical Co., Ltd., Sayama, Japan): 1 g of this product contains 1 g of Japanese Pharmacopoeia acetylsalicylic acid.Acetylsalicylic acid mini-tablets (3 mm in diameter): Each tablet contained 10 mg of Japanese Pharmacopoeia acetylsalicylic acid. The test formulation was prepared in-house at the Pharmacy Department of Showa University Hospital.

#### 2.5.2. Administration of Study Formulations

The participants fasted from 9:00 p.m. on the day before drug administration for the first and second periods. During each period, the study drugs (standard preparation and test formulation) were administered with 150 mL of water early in the morning under fasting conditions. After administration of the test drug, the participants were not allowed to eat or drink for 4 h.

#### 2.5.3. Observations and Blood Collection

Several examinations, observations, and measurements were conducted on the day of drug administration to evaluate the effects of the drug and ensure its safety. The schedules for these evaluations are presented in Table 1.

The investigators examined the presence or absence of subjective symptoms or adverse events as required throughout the study.

Blood samples were collected 10 times: 1 time before administration and 1, 2, 3, 4, 5, 6, 8, 10, and 12 h after administration. Five milliliters of blood were collected from the median cubital vein of each participant and immediately cooled and centrifuged (3000 rpm, 10 min) to obtain plasma, which was then frozen at ≤−80 °C until drug concentrations were determined.

### 2.6. Measurement of Blood Drug Concentration

Plasma salicylic acid and acetylsalicylic acid concentrations were measured using high-performance liquid chromatography (HPLC). This analysis was conducted as described by Kees et al. [13]. A column (Tskgel ODS-80Ts; inner diameter 4.0 mm, length 15 cm) was used for the HPLC analysis; the column temperature was set to 30 °C, and detection was performed at a measurement wavelength of 287 mm.

Water/85% phosphoric acid/acetonitrile (740 mL; 900 μL; 180 mL, *v*/*v*) was used as the mobile phase (flow rate, 1.5 mL/min; injection volume, 20 μL).

Before the measurement of blood drug concentration, acetylsalicylic acid, salicylic acid, and 4-methylbenzoic acid (internal standard) were eluted in this order to confirm that they were collected at a certain efficiency. Retention times were approximately 7, 11, and 15 min, respectively. Known concentrations (2, 5, and 10 μg/mL) of acetylsalicylic acid and salicylic acid were added to serum, extracted using the above method (*n* = 2), and then analyzed using HPLC, as described above. The concentration before extraction was plotted against the peak area, linearity and recovery at a constant efficiency was confirmed, and the blood drug concentration was determined.

### 2.7. Calculation of Pharmacokinetic Parameters

Pharmacokinetic parameters were calculated as follows: area under the blood drug concentration–time curve (AUC_0–12_) and maximum blood concentration (C_max_) up to the last blood collection time (12 h after administration) as primary endpoints, and time to maximum blood concentration (t_max_) and half-life to disappearance (t_1/2_) as secondary endpoints.

AUC_0–12_ was calculated using the trapezoidal method based on actual measurements, and the maximum measured value was used as C_max_. The t_1/2_ values were calculated using Microsoft Excel.

### 2.8. Statistical Analysis

#### 2.8.1. Methods of Statistical Analysis

The results were analyzed for all study participants, excluding those who missed endpoint measurement, using data from the first period (dispersion) as a control and comparing the change in the data obtained in the second period (mini-tablets) between the two groups. JMP^®^ Pro 15.0.0 (SAS, Tokyo, Japan) was used for the analysis.

#### 2.8.2. Primary Endpoint

Plasma AUC_0–12_ and C_max_ of salicylic acid were analyzed using analysis of variance (ANOVA), using the values measured in the first period as controls. The significance level was set at 0.05, and 95% confidence intervals were evaluated for AUC_0–12_ and C_max_ in the first and second periods.

#### 2.8.3. Secondary Endpoints

The pharmacokinetic parameters (AUC_0–12_, C_max_) of acetylsalicylic acid in the plasma were calculated. The evaluation parameters, AUC_0–12_, were calculated using the trapezoidal method, and C_max_ was measured.

The pharmacokinetic parameters (t_1/2_ and t_max_) of salicylic acid and acetylsalicylic acid in the plasma were compared between the two groups. For the pharmacokinetic parameters (AUC_0–12_, C_max_) of salicylic acid in plasma, values for each period were analyzed using ANOVA, with the values measured in period 1 as controls. The significance level was set at *p* < 0.05.

#### 2.8.4. Safety Evaluation

Any subjective symptoms or other findings before or after administration of the study drug, or any abnormal vital signs, were considered adverse events, and the degree and relevance to the study drug were determined.

## 3. Results

### 3.1. Dissolution Test

The results of the dissolution test of acetylsalicylic acid mini-tablets and powder are shown in Figure 2.

### 3.2. Subjects

The medication study included six participants, none of whom stopped or dropped out of the study. The study participants were 28.3 ± 6.3 years old, 175.4 ± 5.2 cm tall, 76.4 ± 8.3 kg, and had a body mass index (BMI) of 24.8 ± 2.17.

### 3.3. Blood Concentration

The average concentration-time trends of salicylic acid and acetylsalicylic acid in the blood during Period 1 (powder) and Period 2 (mini-tablets) are shown in Figure 3 and Figure 4, respectively. AUC_0–12_, C_max_, t_max_, and t_1/2_ values for each blood concentration are shown in Table 2.

### 3.4. Safety Endpoints

No adverse events were observed throughout the study period. There were no abnormalities in the subjective symptoms or other findings.

## 4. Discussion

Personalized medicine is gaining increasing significance in modern drug treatment. The promotion of personalized medicine is expected to benefit medical care and health economics in terms of improved efficacy and safety of drugs, improved patient quality of life through the promotion of preventive medicine, and reduced medical costs through more efficient medical care [14]. The target pediatric patient population ranges from preterm neonates and young children to adolescents. However, a full-scale endeavor is underway to address the specific requirements of children through the development of drugs, preparations, and formulations tailored to their needs [3].

Mini-tablets have garnered interest as a novel form of solid medication for children. They employ the same manufacturing techniques, equipment, and formulation as their adult counterparts and are less prone to taste or texture issues [15]. Klingmann et al. compared the acceptability and ability to swallow a syrup formulation and an uncoated mini-tablet (2 mm diameter) in 306 children aged 6 months to 6 years and found that the mini-tablet was superior [16]. Similarly, Spomer et al. evaluated the acceptability and ability to swallow uncoated mini-tablets (2 mm diameter) in children aged 6 months to 6 years and found that mini-tablets are a promising alternative to liquid formulations [17]. Klingmann et al. also evaluated the acceptability of mini-tablets in neonates (2–28 days old) and found that uncoated mini-tablets were an effective alternative to syrup formulations [18]. Furthermore, Klingmann et al. reported that in children aged 6 months and older, administration of 25 or more mini-tablets was well tolerated, feasible, safe, and superior to equivalent doses of syrup [19], suggesting the possibility that mini-tablets could be used for oral medications at higher doses. The results of these studies support the transition from syrups to small-solid dosage forms for children of all ages, as advocated by the WHO [15].

The acetylsalicylic acid used in this study is an antipyretic analgesic prescribed for pediatric patients; it exhibits antiplatelet effects by acetylating and inhibiting cyclooxygenase-1 and inhibiting thromboxane A_2_ production, which promotes platelet aggregation [20]. It has high efficacy in treating [21,22,23,24,25,26,27,28] ischemic heart disease in adults and health insurance coverage for Kawasaki disease, including cardiovascular sequelae in children. Acetylsalicylic acid requires long-term continuous oral administration; in clinical practice, nurses and physicians in pediatric wards sometimes consult with pharmacists about the difficulty in administering acetylsalicylic acid because it is “sour”. Furthermore, poor medication adherence is a problem in the affected children. The information form of acetylsalicylic acid describes its physicochemical properties as “white crystals or powder, with no odor and a slight sour taste” [29]. Acetylsalicylic acid gradually hydrolyzes into salicylic acid and acetic acid in moist air; therefore, the strong, unique acetylsalicylic acid taste is assumed to be due to the hydrolysis of the product dispensed at medical institutions or during storage at home. The adhesion of acetylsalicylic acid powder to the oral cavity makes it especially difficult for infants to swallow without leaving any residue; therefore, some of the powder may remain even when taken with water, resulting in a strong bitter or sour taste. In the clinical drug treatment setting, the unpleasant taste of acetylsalicylic acid powder may prevent children with Kawasaki disease from easily taking it, causing difficulties in drug treatment; however, in-hospital mini-tablet formulations of acetylsalicylic acid are expected to effectively remedy this problem.

Mini-tablets mask the taste of the drug and are a suitable dosage form modification to improve the dose of acetylsalicylic acid powder. To address the problem of children with Kawasaki disease having difficulty taking acetylsalicylic acid, mini-tablets with a diameter of 3 mm were developed from an acetylsalicylic acid powder approved by the Pharmaceutical Affairs Law.

Prior to human administration, in vitro quality evaluation studies were conducted. The disintegration time of the mini-tablets in the JP18 1st fluid (pH 1.2) in this study was 5 min. Therefore, the likelihood of acetylsalicylic acid disintegrating in the oral cavity, dissolving the drug substance, and causing a taste was very low, and the drug formulation was considered capable of masking unpleasant tastes.

The dissolution studies of the formulated acetylsalicylic acid mini tablets were then confirmed. The similarity between the acetylsalicylic acid powder and the mini-tablets was determined following the “Guidelines for Bioequivalence Studies of Generic Drugs.” When the average dissolution of the reference product is ˂85% within 30 min, the results meet one of the following criteria: when the average dissolution of the reference product reaches 85% within the testing time specified, the average dissolution of the test products is within ±15% of that of the reference product at two appropriate time points when the average dissolution of the reference product is around 40% and 85%, or the f2 value is not less than 42. Upon comparing the dissolution profile of acetylsalicylic acid mini-tablets with that of the powder, it was determined that they were equivalent.

Finally, to evaluate the bioequivalence of acetylsalicylic acid mini-tablets, a two-drug, two-phase, open-label crossover study was conducted on healthy adults at a single institution. The drug plasma concentrations of acetylsalicylic acid powder were compared with those in a previous study, in which the pharmacokinetic parameters of a single dose (100 mg) of acetylsalicylic acid administrated orally in healthy adults were as follows: plasma salicylic acid, AUC_0–∞_: 14.6 mg-h/L, C_max_: 4.19 mg/L, t_max_: 1.00 h, and t_1/2_: 1.88 h; plasma acetylsalicylic acid, AUC_0–12_: 0.88 mg-h/L and C_max_: 1.01 mg/L [30]. The AUC and C_max_ values of salicylic acid in the control and intervention groups in this study were similar to those obtained in the previous study [30], suggesting that similar drug effects can be expected. The absorption from the intestine is faster, and the effect is quicker in a powder formulation than in a tablet or capsule formulation [31]. This is because tablets and capsules require more time to disintegrate in the digestive tract. The drug concentration–time curve of the mini-tablets used in this study overlapped with that of the powder. The rates of mini-tablet disintegration, dissolution in the gastrointestinal tract, and absorption of acetylsalicylic acid from the upper duodenum were equal to that of powder absorption.

The additives used in the mini-tablets are widely available, and applying this technology to other medicines could be a significant advancement in addressing the problem of children having difficulty taking their medicine. The results of this study suggest that acetylsalicylic acid mini-tablets can improve adherence in children who have difficulty taking commonly prescribed acetylsalicylic acid powder. The in-hospital formulated mini-tablets are expected to contribute to effective pediatric drug therapy; however, given that the study participants were healthy adults, the values obtained may differ from those obtained in children because their liver and kidney functions are less developed than those of adults. The drug distribution volume may also differ because of differences in body water content. Future studies on pharmacokinetics in children should be conducted using physiologically based pharmacokinetic (PBPK) methodology. Further investigation based on real-life clinical experience is anticipated.

## Figures and Tables

**Figure 1 pharmaceutics-15-02079-f001:**
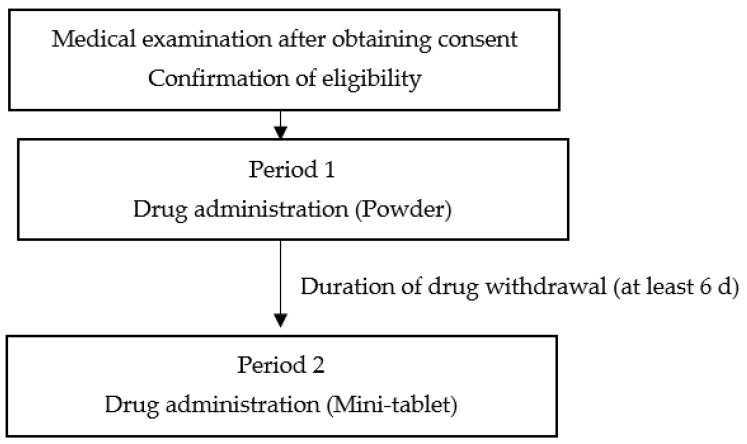
Schedule of the entire study.

**Figure 2 pharmaceutics-15-02079-f002:**
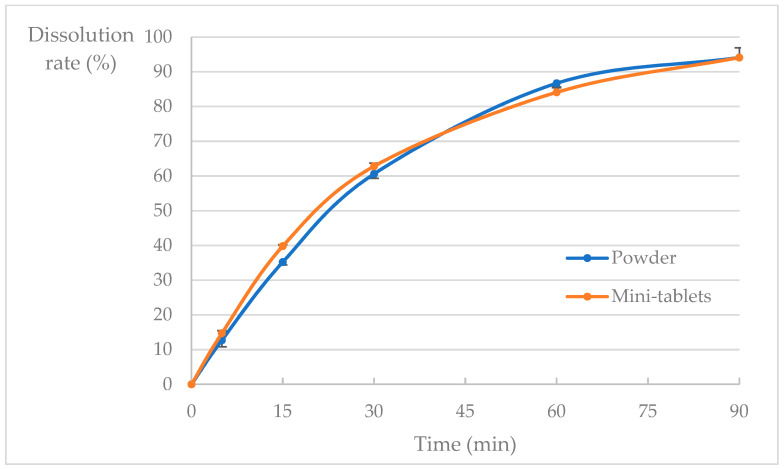
Dissolution rate of acetylsalicylic acid mini-tablets and powder (minimum to maximum values for each lot of 6 vessels at a time; 2 measurements per lot; mean ± standard deviation [SD], *n* = 12).

**Figure 3 pharmaceutics-15-02079-f003:**
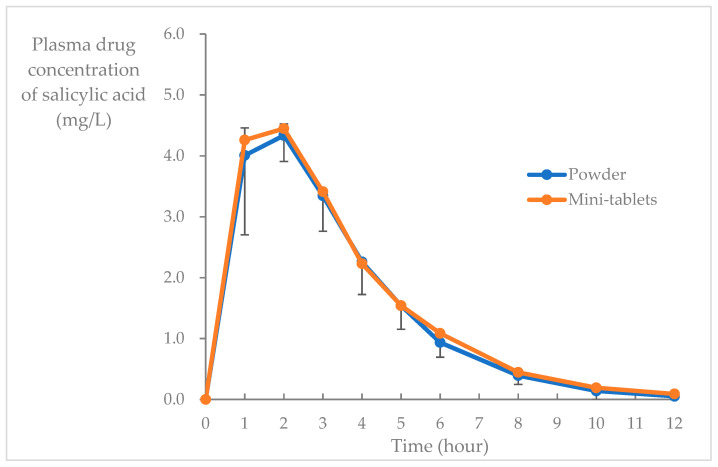
Blood salicylic acid concentration after a single fasting administration of acetylsalicylic acid mini-tablets and powder (tested in different participants; mean ± SD, *n* = 6).

**Figure 4 pharmaceutics-15-02079-f004:**
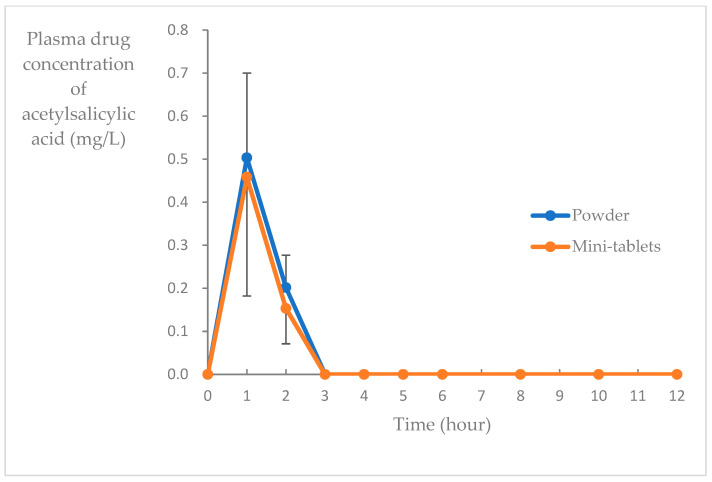
Blood acetylsalicylic acid concentration after a single fasting administration of acetylsalicylic acid mini-tablets and powder (tested in different participants; mean ± SD, *n* = 6).

**Table 1 pharmaceutics-15-02079-t001:** Schedule for research implementation.

Time	Before Administration	9:00	10:00	11:00	12:00	13:00	14:00	15:00	17:00	19:00	21:00
Elapsed time (h)		0	1	2	3	4	5	6	8	10	12
Vital signs measurement	X					X				X	X
Medical examination	X					X					X
Blood sampling	X		X	X	X	X	X	X	X	X	X
Drug administration		X									
Meal						X *				X *	
Subjective symptom survey	X										
Adverse event observation	X										

X in the table indicates the timing of the inspection. * Meals were allowed once blood sampling was completed.

**Table 2 pharmaceutics-15-02079-t002:** Pharmacokinetic parameters.

	Salicylic Acid			Acetylsalicylic Acid		
	1st Period (Powder)	2nd Period (Mini-Tablet)	Difference (2nd Period –1st Period)	1st Period (Powder)	2nd Period (Mini-Tablet)	Difference (2nd Period –1st Period)
C_max_ [mg/L]	4.80 ± 0.79 [0.925–2.074]	5.03 ± 0.97 [4.016–6.049]	0.2313 [−0.898–1.361] *p* = 0.688	0.50 ± 0.20 [0.297–0.710]	0.41 ± 0.24 [0.169–0.664]	−0.093 [−0.293–0.106] *p* = 0.063
AUC_0–12_ [mg-h/L]	18.0 ± 3.03 [14.82–21.19]	18.9 ± 4.59 [14.82–21.19]	0.880 [−1.640–3.400] *p* = 0.563	0.71 ± 0.27 [0.420–0.990]	0.61 ± 0.36 [0.238–0.985]	−0.093 [−0.293–0.106] *p* = 0.313
t_max_ [mg-h/L]	1.67 ± 0.51	1.50 ± 0.54	1.000	-	-	-
t_1/2_ [h]	1.74 ± 0.24	1.86 ± 0.61	0.688	-	-	-

N = 6, mean ± SD, 95% confidence interval, maximum blood concentration (C_max_), area under the concentration-time curve (AUC_0–12_), time to maximum blood concentration (t_max_), vanishing half-reduction period (t_1/2_).

## Data Availability

All relevant data are included in the article.

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
