# Peer review of "A Study on Pharmacokinetics of Acetylsalicylic Acid Mini-Tablets in Healthy Adult Males—Comparison with the Powder Formulation"

_pharmaceutics, 2023, doi:10.3390/pharmaceutics15082079_

Round 1
Reviewer 1 Report
Regarding the manuscript (pharmaceutics-2518293) entitled:
“A Study on Pharmacokinetics of Acetylsalicylic Acid Mini-Tablets in Healthy Adult Males - Comparison with the Powder Formulation”
Comments to the Author
General comment
The manuscript describes an in-hospital mini-tablet formulation of acetylsalicylic acid was developed. Bioequivalence between the newly developed in-hospital mini-tablet and commercial powder formulations cannot be proven by dissolution testing alone to ensure safe and effective use in the treatment of children with Kawasaki disease. I have some few comments to be considered before publication:
1. Abstract: Please data and number in the abstract to provide some insight about the results.
2. Introduction: should be enriched with marked products and formulations in the litrature
3. 2.1.3. Preparation and Storage of Acetylsalicylic Acid Mini-Tablets: A complete characterization of the tablet (assay, friability, hardness, disintegration, …..etc) should be conducted to validate the formulation method and make the Bioequivalence data reproduceable.
4. Figure 2: error bar should be added
Author Response
We wish to express our appreciation to the reviewers for their insightful comments on our paper. The comments have helped us significantly improve the paper.
Point 1: Abstract: Please data and number in the abstract to provide some insight about the results.
Response 1: We wish to thank the reviewer for this comment. We agree that this point requires clarification, and have added a summary of the results to the abstract.
Abstract: Children with Kawasaki disease are prescribed acetylsalicylic acid powder as an antipyretic analgesic and antiplatelet agent. However, some of it to remain in the mouth, leading to a bitter or sour taste. To address this issue, an in-hospital mini-tablet formulation of acetylsalicylic acid was developed. In order to use the mini tablets safely and effectively, dissolution tests alone are not sufficient. Therefore, an open-label crossover study on six healthy participants was conducted to evaluate comparative pharmacokinetic parameters. The pharmacokinetic parameters of salicylic acid were Cmax: 4.80 ± 0.79 mg/L (powder; P), 5.03 ± 0.97 mg/L (mini-tablet; MT), AUC0-12: 18.0 ± 3.03 mg-h/L (P), 18.9 ± 4.59 mg-h/L (MT), those of acetylsalicylic acid Cmax: 0. 50 ± 0.20 mg/L (P), 0.41 ± 0.24 mg/L (MT), AUC0-12: 0.71 ± 0.27 mg-h/L (P), 0.61 ± 0.36 mg-h/L (MT), with no significant differences between the mini-tablet and powder formulations. Although pharmacokinetic results obtained from adults cannot be directly applied to children, the results of this study are important for predicting pharmacokinetics. Furthermore, a formulation that can improve medication adherence in children who have difficulty taking acetylsalicylic acid powder, thus contributing to pediatric drug therapy.
Point 2: Introduction: should be enriched with marked products and formulations in the literature.
Response 2: We agree with the relevance of this reference, and have added it to the Introduction (page 2, line 72 and 87) and References.
Point 3: 2.1.3. Preparation and Storage of Acetylsalicylic Acid Mini-Tablets: A complete characterization of the tablet (assay, friability, hardness, disintegration, …..etc) should be conducted to validate the formulation method and make the Bioequivalence data reproduceable.
Response 3: Thank you for your suggestion. We performed the following tests to determine the characteristics of the tablets. The results are shown below.
Friability (n=6) 0.38%  sd 0.12%
Hardness (n=6) 0.66 kgf  sd 0.10 kgf
Disintegration Time (Purified Water, n=6) 191 sec. sd 28 sec.
Point 4: Figure 2: error bar should be added
Response 4: I’m Sorry the error bars are hard to see. Error bars have already been added but were very small. To make it easier to see, the size of the marker were downsized.
Reviewer 2 Report
The manuscript titled "A Study on Pharmacokinetics of Acetylsalicylic Acid Mini-Tablets in Healthy Adult Males - Comparison with the Powder Formulation" describes the synthesis and characterization of a formulation for pediatric administration of Acetylsalicylic acid. The manuscript is well-written and falls within the topics of Pharmaceutics. The work can be accepted in its current version.
Author Response
Point 1: The manuscript titled "A Study on Pharmacokinetics of Acetylsalicylic Acid Mini-Tablets in Healthy Adult Males - Comparison with the Powder Formulation" describes the synthesis and characterization of a formulation for pediatric administration of Acetylsalicylic acid. The manuscript is well-written and falls within the topics of Pharmaceutics. The work can be accepted in its current version.
Response 1: Thank you very much for taking the time to review my manuscript.
Reviewer 3 Report
The manuscript submitted by Hida et al. is devoted to development of new mini-tablets containing acetylsalicylic acid and investigation of their dissolution properties as well as pharmacokinetics. The study is very well designed and all necessary data were obtained to support the results. I think that the submission is suitable for the publication after making some minor improvements.
1) Page 3, line 108: The described procedure is quite tricky and may turn out to be hard to follow. So, I recommend to enumerate the plastic bags used for preparation of the powder mixture as "plastic bag #1" and "plastic bag #2".
2) Page 3, line 119: Could you please explain what "a punch and die" is? Is this a special detail which has this name? Also, please correct the font here.
3) Page 3, line 124: What was the content of acetylsalicylic acid in the prepared tablets?
4) Page 6, section 2.7: It is not clear, how the PK parameters were calculated - by using only MS Excel or some additional plugin(s) like PKsolver? Is it really possible to calculate the parameters only by "pure" Excel?
5) Page 9, line 305: Please correct the reference in the text: Natalie seems to be the name but not the surname of the scientist.
Author Response
We wish to express our appreciation to the reviewers for their insightful comments on our paper. The comments have helped us significantly improve the paper.

Round 2
Reviewer 1 Report
No comment